# Optical and Electrochemical Properties of a Photosensitive Pyromellitic Diimide Derivative of Cymantrene

**DOI:** 10.3390/molecules28207098

**Published:** 2023-10-15

**Authors:** Elena S. Kelbysheva, Mariam G. Ezernitskaya, Rinat R. Aysin, Tatyana V. Strelkova, Alexey N. Rodionov, Lyudmila N. Telegina

**Affiliations:** A.N. Nesmeyanov Institute of Organoelement Compounds, Russian Academy of Sciences, Vavilova 28, Moscow 119991, Russia; ezernits@mail.ru (M.G.E.); aysin@ineos.ac.ru (R.R.A.); tv.strelkova@yandex.ru (T.V.S.); rodalex@ineos.ac.ru (A.N.R.)

**Keywords:** cymantrene, pyromellitic diimide, photochemistry, optical properties, electrochemistry

## Abstract

Photochemical properties of symmetrical pyromellitic diimide containing two cymantrenyl fragments at two nitrogen atoms were studied with IR, NMR, UV-vis, ESI-MS, and cyclic voltammetry. It was found that new unstable chelates are formed during photolysis. At the same time, the CO ligand dissociates from two Mn(CO)_3_ fragments during photoexcitation, which dramatically changes the electronic and redox properties of the molecule compared to the cymantrene derivative containing one imide fragment. Photolysis leads to a color change from light yellow to green. DFT calculations confirmed the possibility of the formation of complexes due to the loss of one or two CO ligands from manganese atoms. The results obtained with variation of photolysis conditions demonstrated the hemilabile character of the Mn-O=C(imide) bond. On addition of external ligands, the color and electrochemical properties changed, which is promising for the use of this complex as a sensor for small molecules.

## 1. Introduction

Synthesis of organic electro- and photoactive materials necessary for the development and design of modern multiparametric smart materials for organic electronics is an actual task. Rigid covalent organic materials with delocalized π-conjugated systems demonstrate a high degree of charge transfer, which is important for the creation of nonlinear optical materials [1,2,3,4]. Planar π-conjugated aromatic diimides, where electrons migrate rapidly, provide bright radiation in the long-wavelength regions in the form of isolated states with high exciton stability [5,6,7,8]. One of the frequently used templates is pyromellitic diimide, which allows preparation of donor–acceptor compounds with charge transfer of various types by changing the nature of the donor ligand at the N atom of the acceptor diimide fragment. Synthesis of pyromellitic dianhydride derivatives containing pyridylmethyl groups and exhibiting photochromism with tunable photoluminescent properties was reported [9]; compounds with 4-hydroxy acetanilide fragments were synthesized [10], which are multielectron redoxactive compounds. Introduction of 4,6-dimethylpyrimidin-2-yl and 2,3,5,6-tetrafluorophenyl groups allows for tuning the LUMO energy and changing the HOMO-LUMO energy gap [11], whereas synthesis of derivatives with a variety of heterocyclic substituents allows variation of mobility of charge carriers and optoelectronic properties [12]. Proton transfer in the exited state was observed for pyromellitic diimides with benzazole groups, which led to dual fluorescence [13]. The mechanism and binding mode during hierarchical self-assembly of charge transfer complexes using β-cyclodextrin and pyrene- and N-alkyl derivatives of pyromellitic diimides was studied [14]. Pyromellitic dianhydride-based photostable n-channel thin-film transistors with particularly wide energy gaps of 3.56 eV and 3.49 eV in the solid state, respectively, were created [15]. It was also found that the electron mobility in thin-film transistors based on pyromellitic diimides can be controlled with the length of the fluorinated chain at the nitrogen atoms [16]. Bipolar redoxactive organic materials (BROMs) based on N-substituted tetramethylpiperidine 1-oxyl- pyromellitic diimide were developed, which are electroactive compounds having several oxidation states. Due to this unique characteristic, BROM can be used as both an anolyte and catholyte in symmetrical-flow batteries, thus solving cross-contamination problems [17]. Two examples of organometallic material based on pyromellitic diimide are presented in [18,19] concerning synthesis and properties of aromatic diimide carboxylates. On the base of these compounds, novel electrode materials with good capacity retention and rate capability were developed. No other examples of the introduction of an organometallic fragment are available in the literature.

A cymantrenyl (cyclopentadienylmanganesetricarbonyl) fragment is a convenient and available redox group. Earlier, we found that cymantrenylalkylphthalimides have both an oxidation and a reduction potential, which can be varied by changing the solvent nature and substituents in the side chain of the Cp ring and in the phthalimide moiety [20,21]. Photolysis of cymantrenyl derivatives containing n- and π-donor groups in a substituent occurs via CO elimination followed by the formation of chelates with a new hemilabile Mn-donor bond, which allows creation of photochromic systems.

For this reason, synthesis of pyromellitic diimides and investigation of their spectral, electrochemical, and photochemical properties with UV-vis, IR, and NMR methods and cyclic voltammetry were carried out. Comparison of the photophysical and electrochemical properties of these dicarbonyl complexes, as well as the stability and kinetic parameters of ligand exchange with the phthalimides obtained earlier [20], shows how the molecule rigidity and amount of cymantrenyl residues affect. The study of the spatial position of electron donors and acceptors, as well as their number in the molecule, is of great importance and will help to further model materials with specified properties, taking into account both the spatial location of fragments relative to one another and the influence of external factors on them, such as solvents, external ligands, irradiation, or temperature.

## 2. Results and Discussion

### 2.1. Synthesis of N,N′-Di(1-Cymantrenylethyl)Pyromellitic Diimide

Symmetrical diimide with two cymantrenyl fragments (**1**) was prepared according to Figure 1 from pyromellitic dianhydride and aminoethylcymantrene by heating to 100 °C in an Ar atmosphere (Figure 1). Compound **1** was characterized using ^1^H NMR, IR, mass and UV-vis spectra, and also an elemental analysis.

### 2.2. IR Spectroscopic Study of Photochemical and Hemilabile Properties of N,N′-Di(1-Cymantrenylethyl)Pyromellitic Diimide

The IR study of photolysis of **1** was based on the changes observed in the range of stretching vibrations of the MnC≡O and imide groups observed in the course of photolysis and the reverse thermal reaction. The photolysis was performed using irradiation of a solution with the Hg lamp light directly in a sealed IR cell ensuring that released CO remained in the solution.

Earlier, we reported [21] that irradiation of cymantrenyl derivatives, containing a donor substituent in the Cp ring, resulted in chelate formation due to elimination of a CO ligand and coordination of the donor fragment of the substituent to the 16e manganese atom. A similar picture was observed in the case of photolysis of **1**. The IR spectra in the range of metal–carbonyl and imide stretching CO modes of the benzene solution of complex **1** before and after irradiation are presented in Figure 1. In the IR spectra of the benzene solution of the starting compound **1**, two ν(CO) bands (symmetric and degenerate) for the Mn(CO)_3_ moiety at 2024 and 1940cm^−1^ and two bands for the imide fragments (symmetric and asymmetric) at 1772 and 1727 cm^−1^ were observed. After irradiation, the intensities of these bands decreased and new bands appeared as shifted to a low-frequency region compared to the bands of the initial complex **1** (Figure 1), namely, 1938 and 1879 cm^−1^ in the range of ν(MnC≡O) and 1747 and 1648 cm^−1^ in the range of ν(NC=O). This spectral pattern confirms the loss of the CO ligand and coordination of the imide oxygen to 16e manganese to give chelate. It is seen that the intensity of the symmetric ν(CO) band at 2024 cm^−1^ of complex **1** decreases by more than 50%, indicating that both metal tricarbonyl groups participate in chelation.

After photolysis, two new bands at 1747 and 1648 cm^−1^ appear in the spectrum while the bands of the parent **1** remain at 1785 and 1725 cm^−1^. In the presence of CO, the reverse thermal reaction of the photolysis products with CO occurred to give the parent tricarbonyl complex **1**. The reverse reaction is rather slow (half-life time is 181 min and k is 6.38·10^−5^ s^−1^). It is seen from Figure 2 and Appendix A that three structures can be expected for chelates, namely (i) one carbonyl group of one imide fragment is involved in chelation (**2a**), (ii) two carbonyl groups of both imide moieties coordinate to the Mn atom (**2b-*t***) to give *trans* configuration, and (iii) two carbonyl groups of both imide moieties coordinate to the Mn atom (**2b-*c***) to give *cis* configuration. An analysis of the IR spectra in the ν(MnC≡O) and ν(NC=O) regions of complex **1** and its photolysis products allowed us to conclude that (i) the CO ligand is removed from both Mn(CO)_3_ moieties of complex **1** to give chelates, and (ii) one NCO group from each imide fragment coordinates to manganese to give **2b-*t***, **2b-*c***, or their mixture. This conclusion follows from the fact that two new imide bands appear in the spectra of the photolysis product at 1747 and 1650 cm^−1^ associated with symmetric and asymmetric imide modes while the frequencies of non-coordinated (NCO) groups are shifted to 1775 and 1724 cm^−1^, indicating that the kinematic coupling between symmetric and asymmetric vibrations in the chelate decreased due to coordination of one (NCO) fragment, thus increasing the difference between symmetric and asymmetric modes. In the course of the reverse dark reaction, the spectral pattern was gradually restored: the intensities of the chelate bands decreased while the intensities of the bands of **1** increased.

It is not possible to distinguish structures **2b-*t*** and **2b-*c*** (see Appendix A) from IR spectra. In order to confirm the presence of steric isomers in solutions of **1**, the photolysis was performed in a non-polar cyclohexane solution, in which the bands in the IR spectra are narrower and therefore better resolved. Unfortunately, the compounds studied are poorly soluble in cyclohexane, which causes poor quality of the spectra (Figure 2), but it can be seen that both ν(MnCO)s are split, which is evidence for the existence of several isomers arising due to the free rotation of the Mn(CO)_3_ fragments around C-N bonds. The IR spectrum of the photolysis product is presented in Figure 2. In the IR spectrum of the photolysis product (Figure 2, purple line), two ν(MnCO) bands are observed at 1949 and 1897 cm^−1^ characteristic of the chelate, but these bands have a noticeably lesser number of satellites, which speaks for a significantly lower flexibility of the chelate compared to the initial tricarbonyl complex **1**.

We previously reported [20] that some chelates of this type exhibit hemilabile properties in thermal reactions with CO in the presence of an external ligand, for example, CH_3_CN or PPh_3_. In this regard, the photolysis of **1** and the reverse thermal reaction were studied in a benzene solution with addition of CH_3_CN. The course of photolysis was monitored with IR spectra every minute (Figure 3a and Appendix A). After a 1 min irradiation, the intensity of the ν_sym_(CO) band decreased approximately two-fold, and a new band at 1877 cm^−1^ appeared. During further irradiation, this new band gradually shifted to 1873 cm^−1^.

It is seen in Figure 3a that for 4 min, the conversion of **1** is about 80%. At first, dicarbonyl chelates **2** with a ν_asym_(CO) band at 1877 cm^−1^ are formed, which gradually transform to dicarbonyl acetonitrile complexes **3** with a ν_asym_(CO) band at 1873 cm^−1^. This transformation continues even without irradiation. The formation of an acetonitrile complex is confirmed with the IR spectra in the ν_asym_(NCO) range (Figure 3 and Appendix A). It is seen in Figure 3b that after irradiation for 4 min (green line), both type complexes, chelates **2** and acetonitrile complexes **3**, are present in the solution, because their bands of chelates at 1775 and 1748 cm^−1^ are observed, whereas after a 90 min thermal reaction, there are no bands of coordinated chelates and only two bands at 1772 and 1725 cm^−1^ are observed as in the case of starting **1** (Figure 3b). Complex **3** appeared to be quite stable: for 90 min of the thermal reaction, it did not practically transform to complex **1**. The experimental results are supported with DFT calculations (see later).

In the case of photolysis of **1** in the THF solution, two types of complexes are simultaneously formed after irradiation for 4 min, chelates **2** with the Mn-O bond, and a dicarbonyl complex with THF **4** (Appendix A). Two products are formed with two ν_asym_(MCO) bands at 1873 and 1854 cm^−1^; the former coincides with the chelates **2** band and the latter belongs to the dicarbonyl complex with THF **4**. It is remarkable that in the presence of CO, both complexes add CO to form the parent **1**. The isosbestic point observed in the IR spectra indicates that no decomposition takes place in the thermal reaction (Appendix A).

### 2.3. DFT Calculation

The heat effect of photolysis of **1** was calculated with the DFT method. Several products can be expected for this reaction, for example, chelate **2a** and a dicarbonyl complex with the external ligand formed due to the removal of one CO ligand from one cymantrenyl fragment; chelates **2b-*t*** and **2b-*c*** (Appendix A) formed due to splitting of the Mn-CO bond in both cymantrenyl moieties followed by the reaction with acetonitrile to yield dicarbonyl complexes **3b** and **6** (Figure 2 and Appendix A). The energy values calculated at the B3LYP and r2-SCAN-3c levels slightly differ due to different strengths of the chelate Mn…O=C bond and the Mn…NC-Me bond, which are stronger upon introduction of a dispersion correction in the r2-SCAN-3c method. However, the DFT results show that the energies of chelate **2** and the dicarbonyl complex with external ligand **3a** are close, as well as those for compounds **2b-*t,* 2b-*c***, **3b**, and **6** (Appendix A) (ΔG^o^_298_ values vary from 0.7 to 2.6 kcal/mol) (Table 1). Thus, the equilibrium between chelates and the dicarbonyl acetonitrile complex is expected. But since, in the thermal reaction, there is a five-fold excess of acetonitrile, it is shifted toward dicarbonyl complexes **3**, according to the IR data.

### 2.4. Study of Photochemical Properties of N,N′-Di(1-Cymantrenylethyl)Pyromellitic Diimide with ^1^H NMR Spectroscopy and Mass Spectrometry

Unfortunately, poor solubility of the compounds studied limited the efficient use of ^1^H NMR spectroscopy. Nevertheless, it is clearly seen from the ^1^H NMR spectra that after photolysis of **1** during 30 sec, three new signal sets belonging to three reaction products are observed, namely, three overlapping quadruplets at δ 4.76–4.84 ppm from CH protons, which are up-field shifted relative to the signal from the CH proton of the tricarbonyl complex **1** at δ 5.01 ppm. Complete signal assignment does not seem possible due to multiple overlapping of signals.

The irradiation of solution compound **1** in benzene and in benzene with acetonitrile was analyzed using MALDI and ESI mass spectrometry (MS). MALDI is the major method for characterization of poorly soluble imide oligomers [22]. The reaction product is amenable to the analysis of MALDI, preferably in the negative ion mode without a matrix. The spectrum of negative MALDI MS ions for cleavage products in Appendix A shows a series of peaks related to diimide. An ion with *m*/*z* 676 corresponding to the molecular ion of compound **1** is observed in the spectra (Appendix A). Ions with *m*/*z* 592 [M-3CO]^−^ and 508 [M-6CO]^−^ are also identified. Monitoring of the irradiation of benzene solutions showed a change in the ratio of intensity peak [M]^−^ and [M-3CO]^−^ to each other that was observed and no other peak appeared (Appendix A), which is evidence for rapid decomposition of the photolysis products in the course of the recording of the MALDI MS.

The reverse picture was observed in the case of irradiation of **1** in a solvent mixture of benzene/acetonitrile (Appendix A). The basic ion for compound **1** is the ion [M]^−^; the ion [M-3CO]^−^ with *m*/*z* 592 is also present. It means that, unlike the case with the benzene solution, the loss of six CO ligands from two cymantrenyl fragments was not observed. After irradiation, the ion [M-2CO]^−^ with *m*/*z* 620 was detected in the MALDI spectra. Unfortunately, no ions corresponding to dicarbonyl acetonitrile complexes **3** were detected. In all spectra, the ion with *m*/*z* 445 was observed, corresponding to the loss of one cymantrenylethyl fragment to give ion [M-C_5_H_4_CH(CH_3_)Mn(CO)_3_]^−^.

The ESI mass spectra appeared to be more efficient. In contrast to MALDI, we observed a gradual fragmentation of the starting tricarbonyl complex **1**, which is confirmed using the ions with *m*/*z* 676 [M]^−^, 648 [M-CO]^−^, 620 [M-2CO]^−^, etc. MS of a benzene solution after irradiation reveals an increase in the intensities of the ions [M-2CO]^−^ and [M-CO]^−^ relative to the intensity of [M]^−^. Notably, in the course of irradiation, the intensity ratio of [M-2CO]^−^ and [M-CO]^−^ to each other increases, indicating that the amount of chelates **2b** increases compared to **2a** (Figure 4). However, no manganese-containing complexes are observed after a 135 s irradiation (Figure 4c), which is in line with the MALDI MS results.

Monitoring of the irradiation of **1** in a solvent mixture of benzene/acetonitrile showed that the intensity of the ion [M-2CO]^−^ gradually increases compared to [M]^−^ and [M-CO]^−^ with an increase in the irradiation time, indicating that complexes with the loss of two CO ligands are formed; however, no peaks associated with dicarbonyl complexes **3** were observed (5). In contrast to photolysis in benzene, the ions [M-CO]^−^ and [M-2CO]^−^ were present even after 225 s (Figure 5c), which can serve as indirect evidence for the formation of complexes **3**. Apparently, the Mn-NCCH_3_ bond is labile under the conditions of recording MALDI and ESI spectra.

### 2.5. Study of Photochemical and Hemilabile Properties of N,N′-Di(1-Cymantrenylethyl)Pyromellitic Diimide with UV-Vis Spectroscopy

Photochemical and hemilabile properties of **1** and chelates **2** were compared to those of N-(1-cymantrenylethyl)phthalimide **7** (Figure 3) of a similar structure, but having one cymantrenyl and one imide moieties, in order to understand how the molecule rigidity and amount of cymantrenyl residues affect the absorption parameters and kinetic stability. Photochemical properties were studied in different solvents. The corresponding data are presented in Table 2.

The band assignment was performed using the results of the TD-DFT calculations. Remarkably, the UV-vis spectra calculated using the B3LYP functional (Figure 6) better correlate with the experimental results than those obtained using other functionals (Appendix A).

According to the calculation results, the band at 321 nm in the experimental spectra of **1** belongs to the d-d* transition (orbital shapes are given in Appendix A). After photolysis, a new band at 859 nm appears in the spectra (Figure 7a), which refers to the d→π* transition in the formed chelates **2**, i.e., to LMCT. Its high extinction can be explained with the removal of orbital prohibition. Thus, the HOMO (№162) is not pure d, since π-orbitals of the C=O groups of the metal-coordinated pyromellitic fragment make a small contribution. A weak band at 475 nm observed in the spectra is also assigned to LMCT, but the transition occurs to a higher π-orbital. It should be noted that in compound **1** and chelates **2**, transitions LMCT (first eight and nine, respectively) are orbital-shape-forbidden and have a very low intensity; therefore, they are not observed in the spectra. The formation of chelates **2b** should lead to a shift of the LMCT band (Figure 6), but it does not seem possible to distinguish these three chelates in the UV-vis spectra because closely lying wide bands overlap. It is worth noting that ongoing from the phthalimide moiety to the pyromellitic one, the UV-vis spectra significantly change probably due to the fact that the charge transfer upon excitation occurs to the fragment with a larger amount of conjugated π-bonds.

It was found that dicarbonyl chelates **2** and **8** are stable in an inert atmosphere in the absence of CO up to 48 h, but in the presence of CO, the solution color gradually changes to light yellow, which is accompanied by a decrease in the intensity of absorption bands of dicarbonyl chelates and the return of the band of the tricarbonyl complex, thus confirming hemilability of the Mn-O bond in chelates and reversibility of color change (Figure 7b). It is worth noting that the optical densities of both chelate bands at 475 and 859 nm undergo a symbatic decrease (half-life time is 206 min and correlates with IR data (k = 5.61 × 10^−5^ s^−1^)) (Appendix A).

Ongoing from non-polar benzene to polar acetonitrile and DMF, absorption bands of tricarbonyl complexes **1** and **7** are not shifted (Figure 7a; Table 2). However, in the case of **7** in DMF, we observed a bathochromic shift of the band at 293 nm, which is probably related to the solvation effect. Irradiation of **1** in acetonitrile and DMF yields two types of dicarbonyl complexes: complexes with the Mn-solvent bond (**3, 5**, and **9–10**) and chelates **2**. Ongoing from benzene to DMF, a hypsochromic shift of the absorption band by 120 nm was observed (Figure 7b), which confirms the intramolecular charge transfer in chelates because it was reported earlier that for metal-substituted dicarbonyl charge transfer complexes, solvatochromism in the UV-vis spectra was observed [23]. In acetonitrile, the band maximum is shifted only by 10 nm, unlike the case with chelate **9**, for which the corresponding maximum in acetonitrile is shifted by 50 nm (Table 2). Monitoring of the reverse thermal reaction at room temperature with UV-vis spectra showed that the trace amount of the chelates in acetonitrile are observed even after 30 min (Appendix A), while in the case of chelate **9** in acetonitrile, only the bands of the dicarbonyl acetonitrile complex were observed already after 15 min of the thermal reaction [24]. Transformation of chelates **2** to **3** is accompanied by a color change; according to the calculation, a weak absorption band at 366 nm characteristic of the dicarbonyl acetonitrile complex belongs to MLCT. Similar changes were observed earlier for cymantrenylmethylphthalimide [20]. The photolysis of **1** and the subsequent thermal reaction were also performed in benzene with the addition of 10 equiv. acetonitrile; in this case, the rate of ligand exchange decreases and a week band of chelates was observed even after 60 min (Figure 7c). In DMF, the intensity of the bands of chelates **2** decreases during the reverse reaction for 40 min only by 15%. Thus, photoinduced ligand exchange occurs via CO removal from compound **1** to yield chelates **2**, which enter the thermo-induced exchange with acetonitrile to give dicarbonyl complexes **3**. Notably, linking of two (cymantrenylethyl) imide moieties via the benzene ring leads to an increase in kinetic stability of chelates **2** under the conditions of thermo-induced ligand exchange.

### 2.6. Cyclic Voltammetry Study of N,N′-Di(1-Cymantrenylethyl)Pyromellitic Diimide

It is known that the phthalimide group exhibits redox properties; its reduction occurs on two flanking carbonyl groups with the formation of a radical anion localized on the electron-deficient phenyl ring [25,26,27]. The cymantrenyl group is oxidized with one or two electron transfers depending on substituents in the Cp ring [28,29]. The study of electrochemical properties using cyclic voltammetry in DMF showed that the reduction of pyromellitic diimide proceeds at lower potentials compared to that of phthalimide derivative **7** (Figure 8a,c; Table 3), which is probably due to the presence of a rigid bond between the two imide rings. Reduction of **1** is quasi-reversible and occurs at −0.81 V with two electron transfers with formation of a radical anion localized on the CO group of imide fragments (Figure 8c), when compared in the current with the Fc/Fc^+^ standard with a close concentration in the solution. Introduction of cymantrenyl fragments into pyromellitic diimide leads to a decrease in the reduction potential compared to those for organic pyromellitic diimides [27], which indicates that the cymantrenyl group is an acceptor toward the organic fragment [26]. Remarkably, unlike organic analogs [30] where two distinct reduction peaks are observed, in the case of **1**, one broad wave on the reduction branch was recorded, which indicates the proximity of the reduced processes. Also, the reduction proceeds at lower potential values for **1** compared to **7** and electron affinity is increased to 4.96 eV, which speaks to the strong electron-accepting-nature pyromellitic moiety in compound **1** compared to the phthalic one in **7** [26]. Oxidation of Mn(CO)_3_ seems to occur with one electron transfer from each cymantrenyl fragment with formation **1**^2+^, where Mn(CO)_3_ was positively charged. The wave position is significantly shifted towards higher potential values compared to compound **7** (Appendix A).

After irradiation of **1**, two new waves at 0.17 and 0.91 V are observed on the oxidation branch, indicating that upon irradiation of **1** in DMF (Appendix A), two types of complexes are formed, namely, chelates **2** and dicarbonyl complexes **5**, which is in accordance with the results obtained from the UV-vis spectra. The study of the thermal reaction showed that in the absence of CO after 30 min, the peak current of the wave at 0.17 V decreased by about 15%. These data together with an analysis of the UV-vis spectra allow the conclusion to be drawn that oxidation at 0.17 V occurs on the Mn(CO)_2_ moiety of the chelates **2**. On the reduction wave, two distinct quasi-reversible peaks are observed on the reduction branch associated with the reduction of the C=O groups of the imide fragments of dicarbonyl complexes. The first peak appears at a lower voltage than the peak of **1** (Figure 8b). A decrease in the reduction potential value is in line with our previous observations [20] and points to an increase in electron-withdrawing ability of the cymantrenyl fragment ongoing from tricarbonyl to dicarbonyl complexes. The amperage on the waves at −0.81 и −1.59 V decreased insignificantly; therefore, the reduction of both dicarbonyl complexes proceeds simultaneously.

The calculation of electron affinity and ionization energy shows that ongoing from tricarbonyl to dicarbonyl complexes, photoinduced ligand exchange has the most significant effect on the ionization potential, which leads to a significant decrease in the energy gap for dicarbonyl complexes of **2** and **5**. In the case of chelates **2**, Eg determined from CV appeared to be 0.49 eV, which is lower by more than half than that for **5** and **1**. Therefore, electrochemical and electronic properties of substituents on pyromellitic diimide can be tuned by changing the ligand environment around the manganese atom.

## 3. Materials and Methods

^1^H and ^13^C NMR spectra (internal standard solvent, δ from Me_4_Si, DMSO-d_6_—2.50, CDCl_3_—7.26, benzene-d6—7.16 ppm) were recorded on the spectrometer “BrukerAvance 400” (400.13 and 100.61 MHz, respectively). IR spectra were recorded on the Tensor 37 Bruker FTIR spectrometer with a resolution of 2 cm^−1^ in CaF_2_ cells. Electronic absorption spectra were recorded on a Cary 300 spectrophotometer. Photochemical reactions were carried out using an Hg immersion lamp, Hereaus TQ 150, equipped with a special optical glass-cooling jacket, S49. Monochromatic light with a wavelength of 365 nm was isolated by passing the full light of a mercury lamp through a UFS-6 + BS-7 filter. Control over the course of reactions and the purity of products was carried out with the analytical TLC method on Silufol UV-245 (“Kavalier”) plates. Silica gel 60 (Merck) was used for column chromatography. The synthesis of 1-aminoethylcymantrene was described earlier [32].

### 3.1. N,N′-Di(1-Cymantrenylethyl)Pyromellitic Diimide 1

1-Aminoethylcymantrene (2.5 g, 10.1 mmol) was added to a solution of pyromellitic dianhydride (1.0 g, 4.6 mmol) in DMF (25 mL) in an argon atmosphere. The reaction mixture was stirred at 100 °C for 6 h. Then, the mixture was poured into ice water (100 mL) and extracted with CH_2_Cl_2_ (3 × 75 mL). Organic extracts were dried over Na_2_SO_4_; the solvent was removed on a rotary evaporator and **1** was isolated with a yield of 55% (1.7 g). mp 254–257 °C (EtOAc). ^1^H NMR (400 MHz, CDCl_3_, δ ppm, J/Hz) 1.82 (d, 6 H, CH_3_, J = 7.1), 4.62 (m, 2 H, H-Cp), 4.70 (m, 2 H, H-Cp), 4.94 (m, 2 H, H-Cp), 5.19 (q, 2 H, CH, J = 7.5), 5.23 (m, 2 H, H-Cp), 8.26 (s, 2 H, Ph). ^1^H NMR (400 MHz, DMSO-d_6_, δ ppm, J/Hz) 1.70 (d, 6 H, CH_3_, J = 7.1), 4.91 (m, 2 H, H-Cp), 4.96 (m, 2 H, H-Cp), 5.10 (q, 2 H, CH, J = 7.1), 5.21 (m, 2 H, H-Cp), 5.50 (m, 2 H, H-Cp), 8.24 (s, 2 H, Ph). ^1^H NMR (400 MHz, benzene-d_6_, δ ppm, J/Hz) 1.46 (d, 6 H, CH_3_, J = 7.2), 3.72 (m, 2 H, H-Cp), 3.79 (m, 2 H, H-Cp), 4.51 (m, 2 H, H-Cp), 4.62 (m, 2 H, H-Cp), 5.02 (q, 2 H, CH, J = 7.1), 7.76 (s, 2 H, Ph). ^13^C NMR (400 MHz, DMSO-d_6_, δ ppm): 18.7 (CH_3_), 44.5 (CH), 81.1 (Cp), 83.7 (Cp), 85.5 (Cp), 86.8 (Cp), 103.8 (C_i_-Cp), 118.0 (Ar), 137.2 (Ar), 166.0 (C=O), 225.6 (3CO). IR (benzene), ν/cm^-1^: 2024, 1936; 1773, 1729, 1713 (sh). IR (cyclohexane), ν/cm^−1^: 2025, 1946; 1772, 1731, 1696 (sh). Found (%): C, 53.11; H, 2.81; N, 4.17; Mn, 16.3. C_30_H_18_Mn_2_N_2_O_10_. Calc (%): C, 53.28; H, 2.68; N, 4.14; Mn, 16.24.

### 3.2. General Technique of Spectral Studies of Photochemical Reactions of Tricarbonyl Complexes and Their Dark Reaction

Solutions of tricarbonyl **1** in a required solvent (benzene, acetonitrile, cyclohexane, or dimethylformamide with a concentration of 2–4 × 10^−3^ M) were placed in an Ar atmosphere into IR or UV cells and irradiated with a mercury lamp (before irradiation, the lamp was switched on for 2 min) with light at 365 nm at room temperature; spectra were recorded every 0.5–2 min. Irradiation of the samples was carried out up to 50–100% conversion of the tricarbonyl complex. The total irradiation time for all samples was in the range of 0.5–4 min. When obtaining samples for NMR monitoring, solutions of compounds with a concentration of 10–15 × 10^−3^ M were filtered into an NMR ampoule under argon flow, and irradiated with full Hg lamp light at 8–10 °C for 0.5 min to 50% conversion. The distance between the lamp and the sample was 5 cm in all cases. NMR spectra were recorded in 5 mm ampoules. Monitoring of all thermal reactions of the chelates under dark conditions was carried out for 72 h. The procedure “irradiation-dark reaction” for 2 × 10^−3^ M solutions of the tricarbonyl complex in benzene was repeated at least 2 times in IR cells.

### 3.3. Mass Spectroscopic Study

LDI mass spectra were recorded on the Axima Confidence time-of-flight spectrometer (Shimadzu Biotech) in the reflectron high resolution mode with a nitrogen laser (λ = 337 nm). The mass range between 300 and 800 was scanned in the negative ion mode without a matrix for all experiments. The target was prepared with a concentration of a 0.1 mg/mL acetonitrile solution. Resulting mixtures were deposited on a stainless steel target and dried in air. Electrospray ionization mass spectra (ESI MS) of the manganese complexes were registered on a Shimadzu LCMS-2020 High Performance Liquid Chromatograph Mass Spectrometer with the electrospray ionization (ESI) method and single quadrupole detector (negative and positive ions). The desolvation line/heat block temperature were 250/400 °C. Nitrogen (99.5%) was used like a nebulizer and drying gas. Acetonitrile (99.9 + % HPLC gradient grade, Chem-Lab) was used as a mobile phase with a flow rate of 0.2 mL/min without any pretreatment. Complex **1** was dissolved in acetonitrile (injection volume of 1 μL). The mass range between 200 and 800 was scanned.

### 3.4. Electrochemical Study

Electrochemical measurements were performed in standard glass three-electrode equipment using an IPC-PRO potentiostat (Russia) connected to a PC. The working electrode was a glassy carbon disc with an area of 0.03 cm^2^; the counter electrode is a bigger-size platinum plate, and the reference electrode is a saturated silver chloride electrode. The measurements were carried out in DMF containing 0.1 M tetraethylammonium tetrafluoroborate relative to the Fc/Fc+ standard; the potential sweep rate was 100 mV/s. The Fc/Fc+ standard was calibrated at 0.48 V.

### 3.5. Computation Details

Quantum chemical calculations for 1 and its photolysis products were performed using the ORCA v.5 program [33,34]. The thermodynamic ΔHo298 and ΔGo298 parameters were calculated using DFT functional r2-SCAN-3c [35] and B3LYP [36] without correction to harmonic frequencies. Time-dependent (TD) DFT calculations of the first 40 excitations were performed using the B3LYP, r2-SCAN-3c, and PBE0 [37] functional. For all calculations, the Def2-TZVP basis set [38] and RIJCOSX approximation [39,40,41] were used. The expanded integration grids (IntAccX 5,5,5 and GridX 3,3,4) were utilized.

## 4. Conclusions

Photolysis of **1** in different solvents results in formation of several chelate complexes due to removal of one or two CO ligands upon irradiation. It was found that both Mn(CO)_3_ moieties take part in this process. For this reason, several structures can be proposed for chelates **2**, associated with different steric rotamers. It follows from the complicated structure of the IR pattern in the case of the photolysis in cyclohexane solutions. DFT calculations give the close Gibbs energy values for different isomers and allow us to suppose that they are in equilibrium. These chelates in the presence of CO enter into the reverse thermal reaction to give the parent **1,** thus forming a photochromic system. In the UV-vis spectra of **2**, an absorption band at 859 nm is observed, which, bathochromically, shifted by 200 nm compared to the corresponding phthalimide derivative **8** described earlier. In the presence of the external ligand (for example, CH_3_CN), the chelates behave as hemilabile compounds and transform to the dicarbonyl acetonitrile complex. The study of electrochemical properties of pyromellitic tri- and dicarbonyl complexes showed that the cymantrenylalkyl group exhibits electron-withdrawing properties with respect to the pyromellitic diimide fragment and that ligand exchange significantly affects the oxidation potential. Photoinduced ligand exchange at the Mn atom and the introduction of the pyromellitic moiety instead of the phthalimide one allows variation of electronic properties such as the energy gap, thus tuning electrochemical and optical properties.

## Data Availability

Data are available from the authors.

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
