# Peer review of "Optical and Electrochemical Properties of a Photosensitive Pyromellitic Diimide Derivative of Cymantrene"

_molecules, 2023, doi:10.3390/molecules28207098_

Round 1
Reviewer 1 Report
The authors of this manuscript reported on the synthesis, spectral characterization, and investigation of the optical and electrochemical properties of the photosensitive pyromellitic diimide derivative of cymantrene. This work is original, rich in data, and well-written and organized. I have only the following minor comments on this work:
1. Can the authors add to this manuscript (preferably the supporting information) the photo-digital images of the complex before and after the radiation experiments? This will demonstrate clearly to readers the effect of radiation on the color of the complex solution.
2. Was it possible for the researchers to crystallize the complex and obtain its X-ray analysis data?
3. How do the authors confirm that the employed basis sets in their DFT calculations were the optimum ones? Any validation against XRD data for similar complexes was reported previously in the literature?
The English of the manuscript requires very minor editing.
Author Response
- Can the authors add to this manuscript (preferably the supporting information) the photo-digital images of the complex before and after the radiation experiments? This will demonstrate clearly to readers the effect of radiation on the color of the complex solution.
We added
- Was it possible for the researchers to crystallize the complex and obtain its X-ray analysis data?
Unfortunately, we were unable to obtain X-ray analysis data for any of the complexes obtained in the manuscript because these complexes are hemilabile.
- How do the authors confirm that the employed basis sets in their DFT calculations were the optimum ones? Any validation against XRD data for similar complexes was reported previously in the literature?
The basis set Def2-TZVP is so widely used and well validated that it does not require any additional validation. At the same time, the PBE0 functionality is the most universal and reliable functionality ever created; it is against PBE0 that new functionals are tested. Our experience [ doi 10.1016/j.jorganchem.2020.121465, doi 10.1021/acs.inorgchem.6b00572, doi: 10.1016/j.jorganchem.2019.04.01, doi: 10.1021/acs.organomet.9b00434, doi: /10.1021/acs.organomet.0c00351, doi: 10.1016/j.molstruc.2021.130 0 02, doi: 10.1016/j.molstruc.2021.130735, doi: 10.1021/acs.organomet.1c00050, doi: 10.1021/acs.organomet.1c00129] confirms this and allows us to assert that the chosen level of theory is sufficient.
Reviewer 2 Report
In this manuscript, the authors presented a research work that explores the photochemical properties of symmetrical pyromellitic diimide containing two cymantrenyl fragments by IR-NMR, UV-vis, and cyclic voltammetry experiments. Their study revealed that several unstable chelates are formed due to the removal of CO ligands during photolysis in different solvents. The experimental and theoretical investigation of optical and electrochemical properties of these complexes showed their adaptability and tunability, positioning them for diverse applications in materials science and electrochemistry.
The manuscript is well-structured and effectively communicates the research findings. The experimental approach is sound, and the results are presented clearly. The following comments are provided for further improvement:
(1) The manuscript would benefit from a more explicit and robust discussion of the significance and novelty of the research in the introduction section. It is essential to clearly state the importance of the work and point out the novel insights that can be found from the study.
(2) The manuscript and supplementary information contain typographical errors and mistakes. In particular, there are misspellings in the figure captions at several places (for example: SI Figures S24, S25 etc.)
(3) Page 1, line no. 27: Perhaps the authors should provide the full form of 'NLO' for clarity.
(4) Page 4, Figure 2: The caption suggests that the colors of the plots are red and green; however, the actual plots use different colors. Please ensure consistency and accuracy in figure captions throughout the manuscript and supplementary information.
To enhance the manuscript's readability, minor English language editing is required.
Author Response
- The manuscript would benefit from a more explicit and robust discussion of the significance and novelty of the research in the introduction section. It is essential to clearly state the importance of the work and point out the novel insights that can be found from the study.
We changed
- The manuscript and supplementary information contain typographical errors and mistakes. In particular, there are misspellings in the figure captions at several places (for example: SI Figures S24, S25 etc.)
We changed
- Page 1, line no. 27: Perhaps the authors should provide the full form of 'NLO' for clarity.
We provided the full form
- Page 4, Figure 2: The caption suggests that the colors of the plots are red and green; however, the actual plots use different colors. Please ensure consistency and accuracy in figure captions throughout the manuscript and supplementary information.
We changed
Reviewer 3 Report
The photophysical properties of a diimide with two cymantrenyl fragments are studied using several experimental techniques and computational calculations. Light is able to dissociate CO moieties, which bind again in the absence of light. An external ligand can be also linked to the scaffold of the molecule after light irradiation and CO release. All these alterations in the parent molecules cause changes in the photophysical properties that can be interesting for the potential use of these organic species in electronic devices.
Improvement suggestions:
Line 27. What is the meaning of NLO materials?
Line 37. Change the energy gap àchange the HOMO-LUMO energy gap.
Could it be possible to add a scheme summarizing the molecules in the cited references?
Line 66. … in order à …are in order
A band at 165 cm-1 is mentioned in the text (lines 98 and 106). However, a value of 1648 cm-1 is typed in Figure 1.
Line 138. Text mentions a green line in Figure 2, but only one red and one purple lines can be seen in this Figure.
Line 171. Some words seem to be missing
Two consecutive sections are numbered as 2.5
Line 444. 1 should be in bold
Table S1. Provide wider space to the first column.
Figure S24-S26. Could the experimental Vis-UV spectra be added to clearly see which functional is the best to represent it?
Author Response
Line 27. What is the meaning of NLO materials?
We provided the full form
Line 37. Change the energy gap àchange the HOMO-LUMO energy gap.
We changed
Could it be possible to add a scheme summarizing the molecules in the cited references?
We think it is impractical to add a scheme.
Line 66. … in order à …are in order
We changed
A band at 165 cm-1 is mentioned in the text (lines 98 and 106). However, a value of 1648 cm-1 is typed in Figure 1.
We changed
Line 138. Text mentions a green line in Figure 2, but only one red and one purple lines can be seen in this Figure.
We changed
Line 171. Some words seem to be missing
We changed
Two consecutive sections are numbered as 2.5
We changed
Line 444. 1 should be in bold
We changed
Table S1. Provide wider space to the first column.
We changed
Figure S24-S26. Could the experimental Vis-UV spectra be added to clearly see which functional is the best to represent it?
We added